# The Regulatory Functions of Circular RNAs in Digestive System Cancers

**DOI:** 10.3390/cancers12030770

**Published:** 2020-03-24

**Authors:** Xiao Yuan, Ya Yuan, Zhi He, Diyan Li, Bo Zeng, Qingyong Ni, Mingyao Yang, Deying Yang

**Affiliations:** 1College of Animal Science and Technology, Sichuan Agricultural University, Chengdu, Sichuan 611130, China; s20171203@126.com (X.Y.); yuanya145@hotmail.com (Y.Y.); zhihe@sicau.edu.cn (Z.H.); diyanli@sicau.edu.cn (D.L.); apollobovey@163.com (B.Z.); niqingyong@hotmail.com (Q.N.); 2Farm Animal Genetic Resources Exploration and Innovation Key Laboratory of Sichuan Province, Sichuan Agricultural University, Chengdu, Sichuan 611130, China

**Keywords:** circRNAs, function, action mechanism, digestive system cancer

## Abstract

Circular ribonucleic acids (circRNAs), which are a type of covalently closed circular RNA, are receiving increasing attention. An increasing amount of evidence suggests that circRNAs are involved in the biogenesis and development of multiple diseases such as digestive system cancers. Dysregulated circRNAs have been found to act as oncogenes or tumour suppressors in digestive system cancers. Moreover, circRNAs are related to ageing and a wide variety of processes in tumour cells, such as cell apoptosis, invasion, migration, and proliferation. Moreover, circRNAs can perform a remarkable multitude of biological functions, such as regulating splicing or transcription, binding RNA-binding proteins to enable function, acting as microRNA (miRNA) sponges, and undergoing translated into proteins. However, in digestive system cancers, circRNAs function mainly as miRNA sponges. Herein, we summarise the latest research progress on biological functions of circRNAs in digestive system cancers. This review serves as a synopsis of potential therapeutic targets and biological markers for digestive system cancer.

## 1. Introduction

Digestive system (oral cavity, oesophagus, stomach, colorectum, liver, and pancreas)-related cancers seriously affect human health. To date, multiple digestive system cancers have been identified. Oral cancer is a general term for malignant tumours that occur in the mouth, ninety percent of which are oral squamous cell carcinomas (OSCCs) [1]. Globally, a large number of new cases of OSCC are diagnosed every year [2]. Oesophageal cancer, mainly oesophageal squamous cell carcinoma (ESCC), is not only the eighth most common cancer but also the sixth most common cause of death from cancer throughout the world. There are geographical differences in the morbidity and mortality of ESCC. For example, China has a high incidence of oesophageal cancer [3]. Furthermore, gastric cancer (GC) is also commonly diagnosed and is identified as a cause of death from cancer [4]. Colorectal cancer (CRC) is the third and the second most frequently diagnosed cancer in males and females, respectively. In most parts of the world, the CRC incidence ratio is higher in men than in women [5]. Hepatocellular carcinoma (HCC), which is the most common malignant liver tumour, is the second primary cause of cancer-related death globally [6]. In addition, pancreatic cancer (PC) is a highly lethal disease in older adults, for which the mortality closely parallels the incidence [7]. Consequently, a more profound comprehension of the molecular mechanisms of cancer progression in the digestive system is indispensable, which provides new ideas for the identification of new prognostic and diagnostic biomarkers for their prevention and treatment.

The term “circular ribonucleic acids (circRNA)” generally refers to a newly identified class of endogenous noncoding RNAs (ncRNAs). CircRNAs are covalent closed-loop structures without the 5′ caps or 3′ poly (A) tails that linear RNAs possess, which makes them more resistant to RNA exonucleases [8]. Furthermore, as shown in Figure 1A, circRNAs can originate from exons, introns (including intron lariats), or both exons and introns [9]. Importantly, different subclasses of circRNA molecules are produced by distinct biogenesis mechanisms [10,11]. The formation mechanism of exonic circRNAs (ecircRNAs) and exon-intron circRNAs (EIciRNAs) mainly includes three models: exon-skipping or lariat-driven circularisation, direct back-splicing or intron-pairing-driven circularisation, and RNA-binding-protein-driven circularisation [10]. The formation of intronic circRNAs (ciRNAs) depends on 2’,5’-phosphodiester formation from three different pathways [11]. In the 1970s, circRNAs were initially discovered as viroids in RNA viruses [12]. With the continuous improvement of high-throughput sequencing technology and bioinformatics, abundant and diverse circRNAs have been identified in various developmental stages and physiological conditions in numerous organisms, including humans [13], mice [13], and monkeys [14]. Many reports have revealed that circRNAs are indispensable in various physiological and pathological processes [15,16,17,18,19]. Recently, multiple possible biological functions of circRNAs have been discovered (Figure 1B). Typically, circRNAs function as efficient microRNA sponges or competing endogenous RNAs (ceRNAs) to affect gene expression [20]. In addition, circRNAs can regulate transcription [21] and protein functions [22]. Importantly, circRNAs are considered ncRNAs and have coding abilities [23]. In addition, circRNAs have been gradually confirmed to be related to ageing processes and various human diseases, including nervous system disorders, cardiovascular disorders, and cancers (e.g., gastrointestinal cancers) [24,25]. In this article, we highlight recent research on the latest progress in circRNA research on digestive system cancer and gut ageing. Finally, the future of circRNA research progress and directions for the diagnosis and targeted therapy of digestive system cancers are discussed.

## 2. CircRNAs Dysregulated in Digestive System Cancers

With the continuous improvement of high-throughput sequencing technology, numerous differentially expressed circRNAs have been identified in digestive system cancers (Table 1). For example, Sun et al. performed a comparative analysis of the global circRNA expression profiles between the malignantly transformed oesophageal carcinoma cell line SHEEC and the immortalised human oesophageal epithelial cell line SHEE [26]. They identified 813 significantly upregulated and 445 downregulated circRNAs [26]. Through differential gene expression analysis of ESCC and adjacent non-neoplastic tissues, 267 circRNAs—92 upregulated and 175 downregulated—were found to be dysregulated in ESCC tissues compared with non-neoplastic tissues [27]. Similarly, Shi et al. unearthed 744 dysregulated circRNAs in ESCC tissues [28]. The greater number of differentially expressed circRNAs found in the cell lines than in corresponding cancer tissue samples may be due to the differences between the simulated in vitro environment and the in vivo environment. Furthermore, different numbers of dysregulated circRNAs were found in the same kind of samples, possibly because of differences in the RNA quality, sequencing equipment, sequencing reagents, etc. The final results of high-throughput sequencing are affected by several factors, including the sample type, quantity, and quality; the sequencing equipment and reagents; and the subsequent data processing methodology. In conclusion, circRNAs might be implicated in the oncogenesis and evolution of digestive system cancers and could serve as new clinical diagnostic markers and therapeutic targets.

## 3. The Functions of CircRNAs in Digestive Tract Cancers

Recently, increasing evidence has demonstrated that many circRNAs potentially exhibit biological functions in digestive system cancers. Via influencing cell migration, proliferation, apoptosis, and invasion in digestive system cancer, circRNAs act as either oncogenes or tumour suppressors. Some mechanisms have been identified, such as serving as miRNA sponges and regulating protein functions. Among these, the principal mechanism of circRNAs unearthed in digestive system cancers is miRNA sponging. The mechanisms and functions of dysregulated circRNAs in digestive system cancers are displayed in Table 2.

### 3.1. Oral Cancer

CircDOCK1 (hsa_circ_100721) is associated with the apoptosis of OSCC cells, and the circDOCK1-miR-196a-5p-BIRC3 axis has great importance in OSCC [34]. Both circDOCK1 silencing and an increase in the miR-196a-5p mimic level led to increased apoptosis and decreased BIRC3 expression in OSCC cells [34]. In addition, in OSCC cell lines, hsa_circ_0008309 may exert an anti-cancer effect by manipulating the hsa_circ_0008309-miR-382-5p/miR-136-5p-ATXN1 pathway [35], and hsa_circ_0008309 overexpression inhibited miR-382-5p and miR-382-5p expression and increased ATXN1 expression. In summary, circDOCK1 and hsa_circ_0008309 act as tumour suppressors of OSCC and may be latent biomarkers for OSCC (Figure 2a).

### 3.2. Oesophageal Cancer

A previous study discovered that 3 circRNAs (hsa_circ_0067934, circ-TTC17, and circ-DLG1) act as oncogenes in ESCC (Figure 2b) [36,37,38]. In vitro, hsa_circ_0067934 silencing blocked cell cycle progression and inhibited the cell migration and proliferation of ESCC [36]. Similarly, circ-TTC17 knockdown significantly decreased the migration and proliferation of ESCC cells [37]. Furthermore, circ-DLG1 knockdown significantly reduced the proliferation of ESCC cells. Prediction and annotation analyses revealed that circ-DLG1 can sponge 20 miRNAs and affect 60 corresponding target mRNAs [38]. Importantly, 2 circRNAs (hsa_circ_0001946 and circ-SMAD7) were found to act as tumour suppressors in ESCC [37,39]. The overexpression of hsa_circ_0001946 restrained ESCC cell invasion, migration, and proliferation [39]. In addition, hsa_circ_0001946 was verified to predict disease-free survival (DFS), overall survival (OS) and recurrence in FFPE and frozen tissues. Furthermore, circ-SMAD7 knockdown promoted cell migration and proliferation ability, whereas circ-SMAD7 overexpression showed the opposite effect in ESCC [37]. Thus, these findings provide new ideas for screening for ESCC biomarkers.

### 3.3. Gastric Cancer

To date, 11 circRNAs—five upregulated (cancer-promoting) and six downregulated (cancer-suppressing)—have been found to be involved in GC (Figure 2c). The action mechanisms of six of these circRNAs have been described, revealing the pathological process of GC and providing new approaches for the prevention and therapy of GC.

#### 3.3.1. CircRNAs that Act as Oncogenes in GC

By sponging members of the miR-125 family, the oncogenic circRNA circPVT1 may facilitate GC cell proliferation [15]. In addition, circRNA_001569 overexpression promoted cell viability and decreased GC apoptosis by suppressing miR-145 expression, which is mediated by NR4A2 [18]. Furthermore, inhibition of 3 circRNAs (circRNA0047905, circRNA0138960 and circRNA7690-15) restrained the cell invasion and proliferation of GC in vitro [19]. Future research should focus on the mechanism of action of these newly discovered circRNAs in GC.

#### 3.3.2. CircRNAs that Act as Tumour Suppressors in GC

Six circRNAs—circYAP1, circLARP4, circRNA_100269, circPVRL3, circ-ZFR, and circ-104916—have been reported to act as tumour suppressors in GC (Table 2). Previous studies analysed the functional mechanisms of circYAP1, circLARP4, circRNA_100269, and circ-ZFR and found that they are ceRNAs that bind to miRNAs to inhibit GC. For example, circYAP1 restrained the invasion and proliferation of GC cells by binding miR-367-5p to inhibit p27Kip1 expression [40]. In addition, circLARP4 restrained the biological behaviours of GC cells through sponging miR-424 [41]. Furthermore, by targeting the LATS1 gene, miR-424 facilitated the cell invasion and proliferation of GC [41]. CircRNA_100269 was found to target miR-630, suppressing tumour cell growth [42]. In addition, miR-630 could obstruct the function of circRNA_100269 [42]. Circ-ZFR induced cell cycle arrest, suppressed GC cell proliferation, and promoted apoptosis through sponging miR-107/miR-130a and adjusting PTEN, and circ-ZFR regulated the expression level of p53 protein in vivo [43].

Moreover, these circRNAs act through other mechanisms. In particular, circPVRL3 may have protein-coding ability in view of the structure of its open reading frame, internal ribosomal entry sites, and m6A modification [44]. Knockdown of circPVRL3 promoted the cell migration and proliferation of GC [44]. Circ-104916 overexpression in vitro effectively suppressed the cell migration, proliferation and invasion abilities of GC [45]. In addition, circ-104916 downregulated N-cadherin, Vimentin, and Slug and upregulated E-cadherin, which suggests that circ-104916 may participate in epithelial–mesenchymal transition (EMT) [45]. Therefore, overexpression of circRNAs in GC could be an effective method to prevent the progression of GC.

### 3.4. Colorectal Cancer

Ten circRNAs—six upregulated in CRC (cancer-promoting) and four downregulated (cancer-suppressing) circRNAs—have been determined to be involved in the regulation of CRC (Table 2 and Figure 2d). These circRNAs are related to the cell migration, proliferation, and apoptosis of CRC and participate in the regulation of signalling pathways (e.g., the Ras signalling pathway and Wnt/β-catenin pathway). Knowledge about these circRNAs facilitates a deeper understanding of CRC pathogenesis.

#### 3.4.1. CircRNAs that Act as Oncogenes in CRC

Six circRNAs have been found to be upregulated during the progression of CRC, and knockdown of circCCDC66 [46], circRNA-ACAP2 [47], and hsa_circ_0000069 [48] notably restrained CRC cell migration, invasion, and proliferation. In addition, circ-BANP [49] and hsa_circ_0007534 [50] participate in the regulation of CRC cell proliferation. Among these six circRNAs, circCCDC66 [46], circRNA-ACAP2 [47], and hsa_circ_0020397 [51] could be miRNA sponges that regulate the expression level of their targets.

The circRNA circCCDC66 was found to serve as a miRNA sponge to protect MYC mRNA from attack through miR-93 and miRNA-33b [46]. CircRNA-ACAP2 acted as a miRNA sponge to increase the expression of T lymphoma invasion and metastasis protein 1 (Tiam1) through abolishing the inhibitory function of miR-21-5p for Tiam1 expression, thereby promoting the invasion, migration, and proliferation of SW480 colon cancer cells [47]. Moreover, hsa_circ_0020397 inhibited apoptosis and promoted the cell invasion and viability of CRC through enhancing the expression of programmed death-ligand 1 (PD-L1) and telomerase reverse transcriptase (TERT), which are the target genes of miR-138, whereas miR-138 had the opposite effect [51]. Specifically, hsa_circ_0000069 knockdown caused G0/G1 period cell cycle detention in vitro [48]. These findings provide in-depth insights into CRC progression and a reference for the design of therapeutic targets.

#### 3.4.2. CircRNAs that Act as Tumour Suppressors in CRC

Several circRNAs, including circITGA7, cir-ITCH, circ_0026344, and hsa_circ_0000567, have been found to be downregulated in CRC. CircITGA7 acted as a suppressor to repress the cell metastasis and proliferation of CRC by stimulating the transcription of its host gene ITGA7 and inhibiting the Ras signalling pathway [52]. Furthermore, through heightening the level of ITCH and restraining the Wnt/β-catenin pathway, cir-ITCH suppressed the proliferation of CRC cells [53]. In addition, circ_0026344 reduced the cell invasion and growth of CRC while accelerating apoptosis through sponging miR-31 and miR-21 in CRC [54]. Moreover, hsa_circ_0000567 knockdown promoted CRC cell migration and proliferation in vitro [55]. Therefore, overexpression of these circRNAs might regulate CRC.

## 4. CircRNAs Play Important Roles in Digestive Gland Cancers

Digestive gland cancers comprise HCC and PC. Previous studies have reported that circRNAs participate in the regulation of HCC cell migration, invasion, and proliferation and in pancreatic ductal adenocarcinoma (PDAC) cell invasion and proliferation [56,57,58] (Table 3). These circRNAs play a very important role in promoting or inhibiting the growth of digestive gland cancers.

### 4.1. Hepatocellular Carcinoma

A total of 18 circRNAs, 11 of which are upregulated in HCC (cancer-promoting) and the other 7 of which are downregulated (cancer-suppressing), have been found to be involved in HCC (Table 3 and Figure 2e).

#### 4.1.1. CircRNAs that Act as Oncogenes in HCC

Eleven circRNAs have been found to participate in the regulation of HCC cell migration, proliferation, and invasion by two action mechanisms (as ceRNAs and by binding to proteins) (Table 3). CircFBLIM1 [59] and hsa_circ_0078710 [58] promoted HCC cell proliferation, invasion, and migration as ceRNA sponges. CircFBLIM1 was shown to regulate FBLIM1 expression by sponging miR-346 [59]. In addition, in HCC, hsa_circ_0078710 increased HDAC and CDK2 levels by sponging miR-31 and promoted HCC cell cycle progression, invasion, cell proliferation, and migration in vitro, as well as tumour formation by HCC cells in vivo [58]. Cullin 2 (Cul2) circular RNA (circ-10720) and circRNA_104075 have a functional mechanism similar to that of their upstream regulator. A recent study reported that through binding to the Cul2 promoter, Twist1 stimulates its transcription and selectively upregulates Cul2 circular RNA expression. These circRNAs can sponge miRNAs that target Vimentin, thus increasing the expression of Vimentin during EMT [60]. In addition, Cul2 circRNA stimulated the cell invasion, migration and proliferation of HCC [60], and circ_104075 acted as a ceRNA to encourage tumorigenesis by increasing the expression of YAP through sponging miR-582-3p [61]. Mechanistically, HNF4a binds to the -1409 to -1401 region of the circ_104075 promoter to increase the expression of circ_104075.

The remaining circRNAs act as oncogenes in HCC by binding to different miRNAs and influencing different signalling pathways (Table 3), which differentially affect the progression of HCC. Hsa_circ_101280 (hsa_circ_SLAIN1 and hsa_circ_0100929) greatly facilitated HCC tumorigenesis through upregulating JAK2 and adsorbing miR-375, which restrained apoptosis and enhanced proliferation of HCC cells [62]. Hsa_circ_0103809 promoted cell migration and proliferation and inhibited apoptosis in HCC through regulating the miR-490-5p/SOX2 signalling pathway [63]. These findings suggest that circRNAs play a pivotal role in HCC and could help us screen potential biomarkers and targets for the prevention and diagnosis of HCC.

#### 4.1.2. CircRNAs that Act as Tumour Suppressors in HCC

Seven circRNAs have been found to serve as antitumour factors via multiple action mechanisms in HCC. CircARSP91 (hsa_circ_0085154) expression was influenced by the upstream regulator AR-ADAR1 [69]. The androgen receptor (AR) contributes to sex disparity in HCC. AR suppressed circARSP91 expression by upregulating ADAR1 p110 in HCC. In addition, circARSP91 repressed HCC tumour growth both in vivo and in vitro [69]. Similarly, DHX9 could induce downregulation of cSMARCA5 (hsa_circ_0001445) in HCC. cSMARCA5 overexpression facilitated apoptosis and repressed the invasion, migration, and proliferation of HCC cells, and this circRNA upregulated TIMP3 by adsorbing miR-181b-5p and miR-17-3p [70,71]. Furthermore, circMTO1 [72], circADAMTS14 [73], the circRNA SMAD2 [74], and circC3P1 (miR-4641/PCK1) [75] could serve as miRNA sponges to restrain HCC cell tumorigenic behaviours. Moreover, overexpression of both circZKSCAN1 and ZKSCAN1 mRNA repressed the invasion, migration, and proliferation of HCC cells; however, the mechanism of action of circZKSCAN1 was not related to its parental gene ZKSCAN1 (a zinc finger family gene). Overall, circARSP91 and miR-4641/PCK1 could be the most promising biomarkers because of their inhibitory effects on HCC cell migration, invasion, and proliferation and their clear mechanism of action.

### 4.2. Pancreatic Cancer

CircRNA_100782 and circ-PDE8A, which are upregulated in PC, are involved in the regulation of PDAC as miRNA sponges. Chen et al. found that circRNA_100782 knockdown inhibited the proliferation of PDAC cells by decreasing the expression of signal transducer and activator of transcription 3 (STAT3) and interleukin-6 receptor (IL6R), which are the target genes of microRNA-124 (miR-124) [57]. In addition, a study revealed that tumour-released exosomal circ-PDE8A serves as a miRNA sponge for miR-338 to adjust MACC1 and promotes the invasion of PDAC cells through the MACC/MET/ERK or AKT pathways [56]. These findings suggest that circRNA_100782 and circ-PDE8A act as oncogenes in PDAC and could be biomarkers for PDAC (Figure 2f).

## 5. Common CircRNAs Involved in Digestive System Cancers

With increasing research on circRNAs in digestive system cancers, we found that six circRNAs—cerebellar degeneration-related protein 1 antisense RNA (ciRS-7 or Cdr1as), circRNA_100290, circHIPK3, circAGO2, hsa_circ_0014717, and hsa_circ_0001649—are dysregulated and perform similar functions in different cancers (Table 4). In digestive system cancers, the expression of ciRS-7, circRNA_100290, circHIPK3, and circAGO2 is upregulated, while that of hsa_circ_0014717 and hsa_circ_0001649 is downregulated. Among these six common circRNAs, ciRS-7 is involved in four digestive system cancers, namely, ESCC, GC, CRC, and HCC, through sponging miRNA-7 and different targets [77,78,79,80,81]. However, it is unclear how hsa_circ_0001649 affects the progression of CRC or HCC. Next, we explain the functions and mechanisms of these six circRNAs.

### 5.1. CiRS-7

CiRS-7, also called Cdr1as, serves as a super sponge for miR-7 and possesses over 60 common binding sites [20]. The overlapping co-expression of ciRS-7 and miR-7 was observed in the mouse brain, suggesting a high degree of endogenous interaction [91]. A study has found that ciRS-7 may affect the progress of sporadic Alzheimer’s disease (AD) through the miRNA-7/UBE2A signalling circuit [92]. In addition, ciRS-7 has been found to participate in the pathogenesis of multiple tumours (Figure 3A), which demonstrates that ciRS-7 may be a novel therapeutic target and prognostic marker for these deadly diseases.

Recent research revealed that ciRS-7 overexpression in ESCC encouraged cell invasion, migration and proliferation in vitro, as well as lung metastasis and tumour growth in vivo, by abolishing the tumour-suppressive effects of miR-7. Importantly, ciRS-7 served as a sponge for miR-7, resulting in the activation of the NF-κB/p65 pathway mediated by HOXB13 [77]. In addition, knockdown of the stem cell marker Kruppel-like factor-4 (KLF-4), which is a target gene of miR-7, attenuated cell invasion induced by ciRS-7 overexpression. Moreover, BAY 11–7082, an NF-κB inhibitor, also counteracted, to some extent, ciRS-7-mediated cell invasion [78]. Furthermore, ciRS-7 overexpression could block miR-7-induced tumour suppression, including reductions in cell migration and apoptosis, by counteracting the PTEN/PI3K/AKT pathway mediated by miR-7 in GC [79]. CiRS-7 overexpression in CRC cells dramatically reduced the tumour-suppressive activity (repression of cell invasion, migration and proliferation, and acceleration of apoptosis) of miR-7, and ciRS-7 regulated the EGFR/RAF1/MAPK pathway through suppressing the activity of miR-7 [80]. Intriguingly, ciRS-7 silencing inhibited CRC cell invasion and proliferation and decreased IGF-1R and EGFR expression, which could be rescued, in part, through a miR-7 inhibitor [81]. In HCC, ciRS-7 knockdown reduced the expression of PIK3CD and CCNE1, the direct targets of miR-7. In addition, ciRS-7 knockdown restrained HCC cell invasion and proliferation by adsorbing miR-7 [82]. Thus, ciRS-7 has a complex action mechanism in four digestive system cancers and could thus have great potential among the abovementioned six circRNAs as a therapeutic target for digestive system cancers.

### 5.2. CircRNA_100290

CircRNA_100290 (hsa_circRNA_100290) may serve as a ceRNA to adjust the expression of CDK6 by adsorbing miR-29b in OSCC [83] (Figure 3B). Knockdown of circRNA_100290 inhibited the proliferation of OSCC cells in vitro and in vivo [83]. Furthermore, silencing circRNA_100290 greatly decreased the rate of proliferation, restrained the invasion and migration abilities, and encouraged the apoptosis of CRC cells in vitro [84]. CircRNA_100290 is also a ceRNA for FZD4 that sponges miR-516b, resulting in Wnt/β-catenin pathway activation in CRC [84]. Therefore, circRNA_100290 may be an ideal biomarker for OSCC and CRC.

### 5.3. CircHIPK3

CircHIPK3 (hsa_circ_0000284) promoted cell proliferation through serving as a miRNA sponge for miR-124/miR-29b in GC. COL1A1, COL4A1 and CDK6 might play important functions mediated by the circHIPK3-miR-29b/miR-124 axes in GC [85] (Figure 3C). In addition, circHIPK3 knockdown markedly curbed cell invasion, migration, and proliferation, boosted the apoptosis of CRC in vitro, and inhibited CRC metastasis and growth in vivo. Furthermore, circHIPK3 overexpression significantly counteracted the miR-7-mediated repression of malignant phenotypes of CRC cells through upregulating the proto-oncogenes (FAK, IGF1R, EGFR, and YY1) targeted by miR-7. An upstream regulator of circHIPK3 expression in CRC is c-Myb, which is a transcription factor [86]. These findings suggest that circHIPK3 may be a therapeutic biomarker for GC and CRC. In addition, circHIPK3 regulates the proliferation, migration, invasion and apoptosis of non-small-cell lung cancer (NSCLC) cells through miR-149-mediated FOXM1 expression regulation, potentially providing novel insight into the pathogenesis of NSCLC [93].

### 5.4. CircAGO2

CircAGO2 (hsa_circ_0135889), an intronic circRNA produced from the AGO2 gene, was upregulated in both GC and CRC and promoted the invasion, metastasis and cell growth of cancer in vitro and in vivo [87]. In addition, circAGO2 facilitated its enrichment and activation on the 3’-untranslated region of target genes by interacting with human antigen R (HuR) protein, reducing argonaute 2 (AGO2) binding and repressing the silencing of AGO2/miRNA-mediated genes related to cancer progression [87]. Overall, circAGO2 might regulate cancer progression by facilitating the human antigen R (HuR)-inhibited effects of argonaute 2 (AGO2)-miRNA complexes. These results demonstrate that circAGO2 may be a potential therapeutic and diagnostic target for GC and CRC.

### 5.5. Hsa_circ_0014717

The overexpression of hsa_circ_0014717 induced G0/G1 period cell cycle arrest, restrained colony formation and CRC cell proliferation in vitro and inhibited xenograft tumour growth in vivo. Furthermore, hsa_circ_0014717 upregulated the expression of the cell cycle inhibitory protein p16 [94], and the knockdown of p16 reversed the inhibition of hsa_circ_0014717 on CRC cell growth [94]. Shao et al. discovered that hsa_circ_0014717 is stable in human gastric juice and that its characteristics make it suitable for use in clinical detection methodologies [88]. These results suggest that hsa_circ_0014717 may be a useful diagnostic marker for CRC.

### 5.6. Hsa_circ_0001649

Hsa_circ_0001649 overexpression inhibited cell invasion, migration, and proliferation and stimulated apoptosis in HCC [89]. In addition, the expression of hsa_circ_0001649 was heightened in serum samples from patients with CRC after surgery [90]. However, the concrete mechanism of action of hsa_circ_0001649 in HCC and CRC is not clear. Future studies should emphasise the function and mechanism of hsa_circ_0001649 and its potential as a biomarker in HCC and CRC.

## 6. The Clinical Significance of CircRNAs in Digestive System Cancers

### 6.1. Diagnostic Value Assessment

To date, circRNAs have been reported as potential diagnostic targets based on several efficient indicators in five digestive system cancers, such as receiver operating characteristic (ROC) curves, sensitivity and specificity (Table 5).

The diagnostic value of the combined circRNAs is higher than that of the single circRNAs in OSCC and PC. Three circRNAs have been identified as potential biomarkers for the diagnosis of OSCC, including hsa_circ_001242 [95], hsa_circ_0001874 and hsa_circ_0001971 [96]. Notably, the AUC (0.922) of the combination of hsa_circ_0001874 and hsa_circ_0001971 was larger than that of a single circRNA. A similar result was found in PC. Recent research revealed that circ-LDLRAD3 may be a new biomarker in the diagnosis of PC [97]. The expression of circ-LDLRAD3 in PC tissues was correlated with clinical stage, T classification, venous invasion, and lymphatic invasion, and in the plasma of patients with PC, it was significantly associated with CA19-9, N classification, venous invasion, clinical stage, metastasis, and lymphatic invasion [97]. Moreover, the AUC of circ-LDLRAD3 was 0.67. When combined with CA19-9, the AUC was increased to 0.87. These results suggest that the higher diagnostic value of the combination circRNAs could be an important choice for digestive system cancers.

The diagnostic values of circRNAs in different tissues differ in GC and PC. Six circRNAs—hsa_circ_002059 [98], hsa_circ_0000190 [99], hsa_circ_0000745 [100], hsa_circ_0000520 [26], hsa_circ_0000181 [101], and has_circ_0074362 [102]—may be potential novel biomarkers for the diagnosis of GC (Table 5). Of the six circRNAs, plasma hsa_circ_0000520 had the highest AUC (0.8967), sensitivity (0.8235), and specificity (0.8444) [26] and could therefore be a suitable diagnostic target for carcinoembryonic antigen (CEA) of GC patients’ plasma. Furthermore, hsa_circ_0000190 had a relatively higher diagnostic value for the combination of GC cancer tissue and patient plasma with AUC (0.775), sensitivity (0.712), and specificity (0.75) [103]. In PC patients, circRNA_104075 (hsa_circ_104075) in cancer tissues and patient serum had the highest AUC (0.973), sensitivity (0.96), and specificity (0.983) for diagnosis [61]. Thus, circRNAs can be selected as diagnostic targets for different digestive system cancer tissues.

### 6.2. Therapeutic Potential of CircRNAs

CircRNAs could have several advantages as potential targets for the treatment of digestive system cancers. First, circRNAs are stable molecules that may be relatively abundant in cells, tissues, and developmental stages. In addition, circRNAs are resistant to RNase R activity and exist in tissue [101], blood [104], saliva [96], and exosomes [105]. Moreover, circRNAs are highly conserved among species [106]. To date, multiple circRNAs involved in the regulation of digestive system cancer progression have been by in vivo and in vitro experiments [68,107]. A previous study suggested that circRNAs offer new opportunities to design innovative therapeutic strategies for cancer treatment [106]. Thus, circRNAs are very promising as treatment targets. For example, CiRS-7/Cdr1as is a miR sponge that is upregulated and involved in the regulation of ESCC [77], GC [78], CRC [79], and HCC [80] by different pathways, which suggests that the downregulation of ciRS-7/cdr1as expression might provide an effective treatment method. CircRNAs can act as sponges for miRNAs. Artificial circRNA sponges resistant to nuclease digestion could be synthesised using simple enzymatic ligation steps [108]. The synthetic circular miR-21 (scRNA21) sponge inhibited cancer cell proliferation and suppressed the activity of miR-21 on downstream protein targets, including the cancer protein DAXX [108]. Therefore, by studying the structure of endogenous circular RNA sponges to design and develop effective artificial sponges, we could eventually regulate the function of miRNAs in diseases.

## 7. Human Disease-Related CircRNA Databases

Following the application of high-throughput RNA sequencing and various modern computational approaches, a large number of human circRNA-related datasets have been established (Table 6). The circRNA basic information integration and analysis databases are circBase [111], CIRCpedia v2 [112], circView [113], and circBank [114]. It is worth noting that circBase and circBank provide the system nomenclature for circRNAs. However, circBase may fail to achieve accurate classification and positioning. CIRCpedia v2, circView, and circBank achieve more comprehensive and expert basic information integration regarding human circRNA. There are seven databases for the functional annotation of circRNAs (Table 6). The DeepBase v2.0 database focuses on the identification, expression, evolution, and function of circRNAs [115]. The CircInteractome and circNet databases emphasise circRNA-miRNA-gene and circRNA-protein regulatory networks [116]. The circFunBase database specifically provides the experimentally validated and computationally predicted functions of circRNA and circRNA-miRNA interaction networks [117]. The system nomenclature, function, and regulatory networks of human circRNAs (including digestive system cancer-related circRNAs) could be obtained from the above databases.

Moreover, the CSCD [118], circ2Traits [119], exoRBase [104], circRNA disease [120], circR2Disease [121], and somamiR 2.0 [105] databases integrate circRNA-human disease networks. Specifically, the circR2Disease (experimentally supported circular RNAs associated with various diseases) and somamiR 2.0 (miRNA-circRNA network) databases provide circRNA references in GC, CRC, HCC, and PC. The circRNA disease database presents experimentally supported circRNA-disease associations, including GC, CRC, and HCC [120]. Furthermore, a previous study demonstrated that exosomes can modulate the behaviour of recipient cells and may be used as circulating biomarkers for diseases [122]. ExoRBase aids researchers in identifying molecular signatures in blood exosomes and discovering new exosomal biomarkers and functional implications for human diseases, such as CRC, HCC, and PC [105].

In summary, these databases help researchers understand the essential information on circRNAs in digestive system cancers and other human diseases and promote understanding of the occurrence and development of disease as well as its diagnosis and prognosis.

## 8. Techniques and Methods of CircRNA Study in Digestive System Cancers

### 8.1. Identification of CircRNAs

Several techniques have been used to detect circRNAs in digestive system cancers from primary tumours and control tissues or cells [49,60], including microarray analysis [49] and high-throughput RNA sequencing [60]. Microarray analysis is suitable for screening known human circRNAs. High-throughput RNA sequencing is suitable for identifying new circRNAs and splice sites of circRNAs with sufficient underlying sequencing depth. Total RNA was treated with RNase R to degrade the linear RNA in library construction for screening circRNAs [43]. Then, circRNAs were experimentally validated by reverse transcription-quantitative polymerase chain reaction (RT-qPCR) with GAPDH as the reference gene [43]. Covalent loci of circRNA can be verified through Sanger sequencing. Furthermore, bioinformatics analysis is an accurate and convenient approach that can rapidly generate helpful information for further verification [127]. Thus, human disease-related circRNA databases could help us to analyse nomenclature, expression patterns, and evolution and predict the functions of human circRNAs.

### 8.2. Function and Mechanism of CircRNA

To provide the foundation for research on circRNA function, the subcellular location (nucleus or cytoplasm) of circRNAs first need to be analysed based on efficient nuclear/cytoplasmic RNA separation [44] and fluorescence in situ hybridisation (FISH) [48,52]. Then, gain-of-function (GOF) and loss-of-function (LOF) are employed to study circRNA functions in digestive system cancer cell lines and xenograft nude mouse models [46,67,72,75]. Human circRNA can be synthesised and cloned into the overexpression vector pcDNA3.1 [45]. The circRNA-pcDNA3.1 overexpression vector can then be transfected into digestive system cancer cell lines [40]. Next, those transfected cell lines are injected into a xenograft nude mouse model to test the GOF of circRNAs in an animal model [46,67,68,72,73,75]. On the other hand, small interfering RNAs (siRNAs) have been designed for the junction of circRNAs and transfected into digestive system cancer cell lines to examine the LOF of circRNAs [23]. Thus, multiple pathological processes can be researched with the aid of GOF and LOF of circRNAs in digestive system cancer, such as cell migration, proliferation, invasion, and anchorage-independent growth.

CircRNAs can act as a sponge binding to miRNA, thereby reducing the ability of miRNA to target mRNAs. Multiple databases can be used to analyse circRNA-miRNA-mRNA networks, such as circNet [123] and circFunBase [117] for known circRNAs and Targetscan [128] and RNAhybrid [129] for new circRNAs. A dual-luciferase assay (psiCHECK-2 vector) has been used to validate the relationship between circRNA and miRNA [42,51]. RNA immunoprecipitation (RIP) assays for AGO2 (argonaute 2) in digestive system cancer cell lines have been used to investigate the expression levels of endogenous circRNAs and miRNAs pulled down from AGO2-expressing cells by qPCR analysis [41]. Furthermore, Kyoto Encyclopedia of Genes and Genomes (KEGG) and Gene Ontology (GO) analyses were performed to annotate genes targeted with the purpose of studying how circRNAs regulate parental gene expression [130]. On the other hand, circRNAs can bind to proteins and sequester them to particular subcellular locations [60,69]. Several techniques targeting this mechanism have been developed. To date, RNA pull-down assays and RIPs have been utilised to detect circRNA-protein interactions [131]. Following the discovery of new circRNA mechanisms, the development of new technology will help us to employ the function and regulatory mechanisms of circRNAs.

## 9. Conclusions

CircRNAs are a type of endogenous RNA that are stable and resist RNA nuclease digestion. CircRNAs have emerged as important regulators in multiple digestive system cancers. Dysregulated circRNAs were found to act as oncogenes or tumour suppressors in digestive system cancers (Figure 3). In the complex tumour microenvironment, overexpression of the downregulated circRNAs or depletion of the upregulated circRNAs in patients are the main approaches for circRNA-based therapy. According to current research data, the regulatory mechanism of circRNAs in digestive system cancers is very complex, and the underlying molecular mechanisms by which circRNAs cause and inhibit these cancers remain largely ambiguous. CircRNA transcriptome databases of OSCC, ESCC, GC, CRC, HCC, and PDAC have been reported and are convenient sources for selecting targets and biomarkers. The dysregulated circRNAs between digestive system cancer and normal cells/tissues may participate in regulating the inhibition and promotion of these digestive system cancers, suggesting that circRNAs could serve as therapeutic targets and biomarkers for clinical diagnosis. However, the above assumptions continue to present substantial difficulties and enormous challenges in clinical practice, and more studies on the significance, efficiency, generality, security, and reliability of these approaches are needed. Thus, we anticipate that this review will increase the comprehension of the principal functions of many circRNAs and their multiple regulatory hubs in the progression of digestive system cancers.

## Figures and Tables

**Figure 1 cancers-12-00770-f001:**
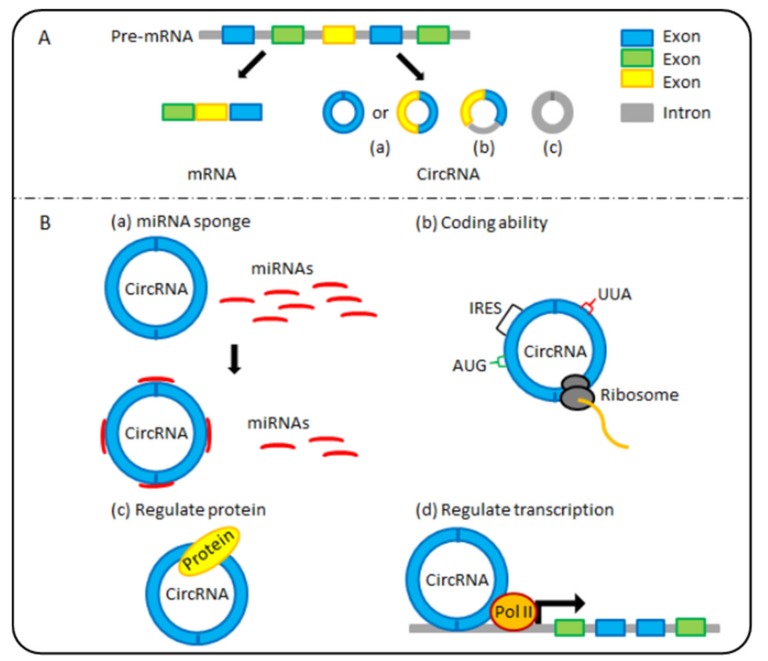
The classification and molecular functions of circular ribonucleic acids (circRNAs). (**A**) According to their sequences, circRNAs are divided into 3 main categories: (a) exonic circRNAs (ecircRNAs), (b) exon-intron circRNAs (EIciRNAs), and (c) intronic circRNAs (ciRNAs). (**B**) CircRNAs have four potential functions: (a) microRNA (miRNA) sponging: some circRNAs serve as efficient miRNA sponges, regulating the activity of miRNA target genes. (b) Coding ability: some circRNAs encode peptides or proteins and affect their biological function. (c) and (d) regulation: circRNAs affect protein function directly and regulate transcription.

**Figure 2 cancers-12-00770-f002:**
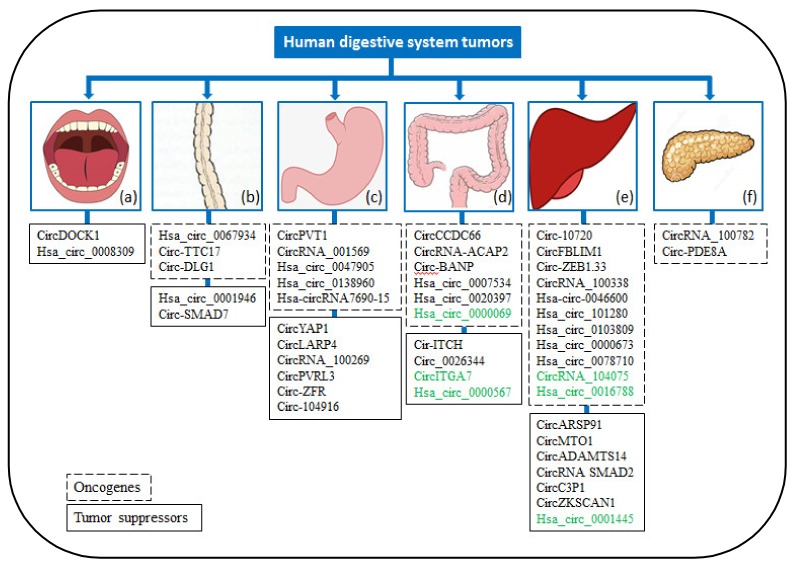
Dysregulated circRNAs act as oncogenes or tumour suppressors and have diagnostic value in digestive system cancers. (**a**) Oral cavity, (**b**) oesophagus, (**c**) stomach, (**d**) colorectum, (**e**) liver, and (**f**) pancreas. Green indicates circRNAs with potential diagnostic value in digestive system cancers.

**Figure 3 cancers-12-00770-f003:**
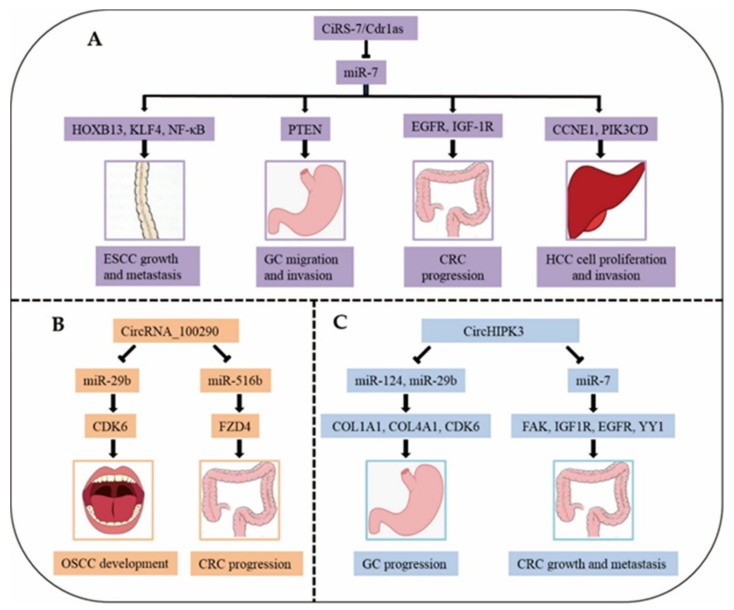
The mechanisms of three circRNAs (CiRS-7/Cdr1as, CircRNA_100290 and CircHIPK3) in different diseases. (**A**), Mechanism of ciRS-7 in ESCC, GC, CRC, and HCC. (**B**), Mechanism of circRNA_100290 in OSCC and CRC. (**C**), Mechanism of circHIPK3 in GC and CRC.

**Table 1 cancers-12-00770-t001:** Overview of differentially expressed circRNAs identified in digestive system cancers.

Cancer Type	Sample	Number of Dysregulated CircRNAs (Up, Down)	References
OSCC	Cancer tissue and adjacent normal tissue	16 (8, 8)	2018 [29]
ESCC	SHEE and SHEEC cell lines	1258 (813, 445)	2017 [26]
ESCC	Cancer tissue and adjacent normal tissue	267 (97, 175)	2017 [27]
ESCC	Cancer tissue and adjacent normal tissue	744 (469, 275)	2018 [28]
GC	Cancer tissue and adjacent normal tissue	713 (191, 522)	2017 [30]
CRC	Cancer tissue and adjacent normal tissue	10245 (6264, 3981)	2018 [31]
HCC	Cancer tissue and adjacent normal tissue	127 (113, 14)	2017 [32]
PDAC	Cancer tissue and adjacent normal tissue	351 (209, 142)	2016 [33]

**Table 2 cancers-12-00770-t002:** CircRNAs in digestive tract cancers.

Cancer Type	CircRNA	Dysregulation	MiRNA Sponge	Target Gene	Function	References
OSCC	CircDOCK1/hsa_circ_100721	Down	miR-196a-5p	BIRC3	Apoptosis (-)	[34]
	Hsa_circ_0008309	Down	miR-136-5p/miR-382-5p	ATXN1	-	[35]
ESCC	Hsa_circ_0067934	Up	-	-	Proliferation (+), migration (+)	[36]
	Circ-TTC17/hsa_circ_0021771	Up	-	-	Proliferation (+), migration (+)	[37]
	Circ-DLG1/hsa_circ_0007203	Up	-	-	Proliferation (+)	[38]
	Hsa_circ_0001946	Down	-	-	Proliferation (−), invasion (−), migration (−)	[39]
	Circ-SMAD7/hsa_circ_0000848	Down	-	-	Proliferation (−), migration (−)	[37]
GC	CircPVT1	Up	miR-125	-	Proliferation (+)	[15]
	CircRNA_001569/hsa_circ_001569	Up	miR-145	NR4A2	Apoptosis (−)	[18]
	Hsa_circ_0047905	Up	-	-	Proliferation (+), invasion (+)	[19]
	Hsa_circ_0138960	Up	-	-	Proliferation (+), invasion (+)	[19]
	Hsa-circRNA7690-15	Up	-	-	Proliferation (+), invasion (+)	[19]
	CircYAP1/has_circ_0002320	Down	miR-367-5p	p27Kip1	Proliferation (−), invasion (−)	[40]
	CircLARP4	Down	miR-424	LATS1	Proliferation (−), invasion (−)	[41]
	CircRNA_100269/has_circ_100269	Down	miR-630	-	Proliferation (−)	[42]
	Circ-ZFR	Down	miR-130a/miR-107	PTEN, p53	Proliferation (−), apoptosis (+)	[43]
	CircPVRL3/has_circ_0066779	Down	-	-	Proliferation (−), migration (−)	[44]
	Circ-104916/has_circ_104916	Down	-	E-cadherin, N-cadherin, Vimentin and Slug	Proliferation (−), invasion (−), migration (−)	[45]
CRC	CircCCDC66	Up	miRNA-33b/miR-93	MYC	Proliferation (+), invasion (+), migration (+)	[46]
	CircRNA-ACAP2	Up	miR-21-5p	Tiam1	Proliferation (+), invasion (+), migration (+)	[47]
	Hsa_circ_0000069	Up	-	-	Proliferation (+), invasion (+), migration (+)	[48]
	Circ-BANP	Up	-	-	Proliferation (+)	[49]
	Hsa_circ_0007534	Up	-	-	Proliferation (+), apoptosis (−)	[50]
	Hsa_circ_0020397	Up	miR-138	TERT, PD-L1	Invasion (+), apoptosis (−)	[51]
	CircITGA7/hsa_circ_0026782	Down	-	Ras signalling pathway, ITGA7	Proliferation (−), migration (−)	[52]
	Cir-ITCH	Down	-	Wnt/β-catenin pathway, ITCH	Proliferation (−)	[53]
	Circ_0026344/hsa_circ_0026344	Down	miR-21/miR-31	-	Proliferation (−), invasion (−)	[54]
	Hsa_circ_0000567	Down	-	-	Proliferation (−), migration (−)	[55]

Note: (+) means promotion and (−) means suppression.

**Table 3 cancers-12-00770-t003:** CircRNAs in digestive gland cancers.

Cancer Type	CircRNA	Dysregulation	Upstream Regulator	MiRNA Sponge	Target Gene	Function	References
HCC	Circ-10720/hsa_circ_10720	Up	Twist1	-	Vimentin	Proliferation (+), invasion (+), migration (+)	[60]
	CircFBLIM1/hsa_circ_0010090	Up	-	miR-346	FBLIM1	Proliferation (+), invasion (+), apoptosis (-)	[59]
	Circ-ZEB1.33	Up	-	miR-200a-3p	CDK6	Proliferation (+)	[64]
	CircRNA_104075/circ_104075	Up	HNF4a	miR-582-3p	YAP	-	[61]
	CircRNA_100338/hsa_circRNA_100338	Up	-	miR-141-3p	-	Migration (+)	[65]
	Hsa-circ-0046600	Up	-	miR-640	HIF-1α	Migration (+)	[66]
	Hsa_circ_0016788	Up	-	miR-486	CDK4	Proliferation (+), invasion (+), apoptosis (−)	[67]
	Hsa_circ_101280/hsa_circ_SLAIN1/hsa_circ_0100929	Up	-	miR-375	JAK2	Proliferation (+), apoptosis (−)	[62]
	Hsa_circ_0103809	Up	-	miR-490-5p	SOX2	Proliferation (+), migration (+), apoptosis (−)	[63]
	Hsa_circ_0000673	Up	-	miR-767-3p	SET	Proliferation (+), invasion (+)	[68]
	Hsa_circ_0078710	Up	-	miR-31	HDAC, CDK2	Proliferation (+), invasion (+), migration (+)	[58]
	CircARSP91/hsa_circ_0085154	Down	AR-ADAR1	-	-	-	[69]
	Hsa_circ_0001445/cSMARCA5	Down	DHX9	miR-17-3p/miR-181b-5p	TIMP3	Proliferation (−), invasion (−), migration (−), apoptosis (+)	[70,71]
	CircMTO1/hsa_circRNA_0007874/hsa_circRNA_104135	Down	-	miR-9	p21	Proliferation (−), invasion (−)	[72]
	CircADAMTS14/hsa_circ_0018665	Down	-	miR-572	RCAN1	Proliferation (−), invasion (−), apoptosis (+)	[73]
	CircRNA SMAD2/hsa_circ_0000847	Down	-	miR-629	-	Invasion (−), migration (−)	[74]
	CircC3P1	Down	-	miR-4641	PCK1	Proliferation (−), invasion (−), migration (−)	[75]
	CircZKSCAN1/hsa_circ_0001727	Down	-	-	-	Proliferation (−), invasion (−), migration (−)	[76]
PDAC	CircRNA_100782/hsa_circ_100782	Up	-	miR-124	IL6, STAT3	Proliferation (+)	[57]
	Circ-PDE8A/hsa_circ_0036627	Up	-	miR-338	MACC1, MET, ERK, AKT	Invasion (+)	[56]

Note: (+) means promotion and (−) means suppression.

**Table 4 cancers-12-00770-t004:** Common circRNAs in digestive system cancers.

CircRNA	Cancer Type	Dysregulation	Upstream Regulator	MiRNA Sponge	Target Gene	Function	References
CiRS-7/Cdr1as	ESCC	Up	-	miR-7	HOXB13, NF-κB/p65	Proliferation (+), migration (+), invasion (+)	[77]
					KLF4, NF-κB	Migration (+), invasion (+)	[78]
	GC	Up	-	miR-7	PTEN/PI3K/AKT	Migration (+), apoptosis (-)	[79]
	CRC	Up	-	miR-7	EGFR/RAF1/MAPK	Proliferation (+), migration (+), invasion (+), apoptosis (−)	[80]
					EGFR, IGF-1R	Proliferation (+), invasion (+)	[81]
	HCC	Up	-	miR-7	CCNE1, PIK3CD	Proliferation (+), invasion (+)	[82]
CircRNA_100290/hsa_circRNA_100290	OSCC	Up	-	miR-29b	CDK6	Proliferation (+)	[83]
	CRC	Up	-	miR-516b	FZD4, Wnt/β-catenin	Proliferation (+), migration (+), invasion (+), apoptosis (−)	[84]
CircHIPK3/hsa_circ_0000284	GC	Up	-	miR-124/miR-29b	COL1A1, COL4A1, CDK6	Proliferation (+)	[85]
	CRC	Up	c-Myb	miR-7	FAK, IGF1R, EGFR, YY1	Proliferation (+), migration (+), invasion (+), apoptosis (−)	[86]
CircAGO2/hsa_circ_0135889	GC/CRC	Up	-	-	HuR	Proliferation (+), invasion (+), migration (+)	[87]
Hsa_circ_0014717	GC	Down	-	-	-	-	[35]
	CRC	Down	-	-	p16	Proliferation (−)	[88]
Hsa_circ_0001649	CRC	Down	-	-	-	-	[89]
	HCC	Down	-	-	-	Proliferation (−), migration (−), invasion (−), apoptosis (+)	[90]

Note: (+) means promotion and (−) means suppression.

**Table 5 cancers-12-00770-t005:** The diagnostic value of circRNAs in digestive system cancers.

Cancer Type	CircRNA	Dysregulation	Sample	Clinicopathological Factors	AUC	Sensitivity	Specificity	References
OSCC	Hsa_circ_001242	Down	Cancer tissue	Tumour size, T stage	0.784	0.725	0.775	[95]
	Hsa_circ_0001874	Up	Patients’ saliva	TNM stage, tumour grade	0.863	0.7444	0.9024	[96]
	Hsa_circ_0001971	Up	Patients’ saliva	TNM stage	0.845	0.7556	0.878	[96]
	Hsa_circ_0001874 + hsa_circ_0001971	-	-	-	0.922	0.9268	0.7778	[96]
GC	Hsa_circ_002059	Down	Cancer tissue	Distal metastasis, TNM stage, gender, age	0.73	0.81	0.62	[98]
	Hsa_circ_0000190	Down	Cancer tissue	Tumour diameter, lymphatic metastasis, distal metastasis, TNM stage, CA19-9	0.75	0.721	0.683	[103]
			Patients’ plasma	CEA	0.6	0.414	0.875	
			Cancer tissue + Patients’ plasma	-	0.775	0.712	0.75	
	Hsa_circ_0000745	Down	Cancer tissue	Tumour differentiation	-	-	-	[100]
			Patients’ plasma	TNM stage	0.683	0.855	0.45	
			Patients’ plasma + CEA	-	0.775	0.8	0.633	
	Hsa_circ_0000520	Down	Cancer tissue	TNM stage	0.6129	0.5357	0.8571	[9]
			Patients’ plasma	CEA	0.8967	0.8235	0.8444	
	Hsa_circ_0000181	Down	Cancer tissue	Tumour diameter, lymphatic metastasis, distal metastasis, CA19-9	0.756	0.539	0.852	[101]
			Patients’ plasma	Differentiation, CEA	0.582	0.99	0.206	
	Hsa_circ_0074362	Down	Cancer tissue	Lymphatic metastasis, CA19–9	0.63	0.362	0.843	[102]
CRC	Hsa_circ_001988	Down	Cancer tissue	Differentiation, perineural invasion	0.788	0.68	0.73	[109]
	Hsa_circ_0003906	Down	Cancer tissue	Lymphatic metastasis, differentiation	0.818	0.803	0.725	[45]
	Hsa_circ_0000567	Down	Cancer tissue	Tumour size, lymph metastasis, distal metastasis, TNM stage	0.8653	0.8333	0.7647	[55]
	Hsa_circ_0026782	Down	Cancer tissue	Tumour size, lymph metastasis, distant metastasis, TNM stage	0.8791	0.9275	0.6667	[52]
	Hsa_circ_0000069	Up	Cancer tissue	TNM stage, age	-	-	-	[48]
HCC	Hsa_circ_0003570	Down	Cancer tissue	Tumour diameter, differentiation, microvascular invasion, BCLC stages, TNM stages, serum AFP	0.7	0.449	0.868	[110]
	Hsa_circ_0001445	Down	Cancer tissue	The number of tumour foci	-	-	-	[71]
			Patients’ plasma	Serum AFP	0.862	0.712	0.942	
	Hsa_circ_104075	Up	Cancer tissue	-	-	-	-	[61]
			Patients’ serum	-	0.973	0.96	0.983	
	Hsa_circ_0016788	Up	Cancer tissue	-	0.851	-	-	[67]
PC	Circ-LDLRAD3	Up	Cancer tissue	Venous invasion, lymphatic invasion, clinical stage, T classification	0.67	0.5738	0.7049	[97]
			Patients’ plasma	CA19-9, N classification, venous invasion, lymphatic invasion, clinical stage, metastasis				
	Circ-LDLRAD3 + CA19-9	-	-	-	0.87	0.8033	0.9355	[97]

**Table 6 cancers-12-00770-t006:** Human disease circRNA-related databases.

Name	Website	Description	References
**CircRNA basic information integration and analysis databases**			
CircBase	http://www.circbase.org/	CircRNA database for six species (human, mouse, worm, *Latimeria chalumnae, Latimeria menadoensis*)	[111]
CIRCpedia v2	http://www.picb.ac.cn/rnomics/circpedia	Comprehensive circRNA annotation from over 180 RNA-seq datasets across six different species (human, mouse, rat, zebrafish, fly, worm)	[112]
CircView	http://gb.whu.edu.cn/CircView/	A user-friendly visualisation tool for circRNAs detected by existing tools and regulatory elements, such as microRNA response elements and RNA-binding protein binding sites (human, mouse, zebrafish, fly, worm)	[113]
CircBank	http://www.circbank.cn/	A comprehensive database for human circRNA with standard nomenclature, miRNA binding sites, conservation of circRNAs, m6A modification of circRNAs, mutation of circRNAs and protein-coding potential of circRNAs.	[114]
**CircRNA-related function analysis database**			
DeepBase v2.0	https://www.rna-seqblog.com/deepbase-v2-0-identification-expression-evolution-and-function-of-small-rnas-lncrnas-and-circular-rnas-from-deep-sequencing-data/	Identification, expression, evolution and function of small RNAs, lncRNAs and circular RNAs from deep-sequencing data across 19 species	[115]
CircInteractome	https://omictools.com/circinteractome-tool	Exploring circular RNAs and their interacting proteins and microRNAs, identifying potential circRNAs which can act as RBP sponges, designing junction-spanning primers for specific detection of circRNAs of interest, siRNAs for circRNA silencing, and potential internal ribosomal entry sites (IRES).	[116]
CircNet	http://circnet.mbc.nctu.edu.tw/	Providing the following resources: novel circRNAs, integrated miRNA-target networks, expression profiles of circRNA isoforms, genomic annotations of circRNA isoforms, sequences of circRNA isoforms, tissue-specific circRNA expression profiles, and circRNA-miRNA-gene regulatory networks	[123]
CircRNADb	http://reprod.njmu.edu.cn/circrnadb	Providing genomic information, exon splicing, genome sequences, internal ribosome entry sites (IRES), open reading frames (ORF) and references	[124]
TSCD	http://gb.whu.edu.cn/TSCD	Comprehensive characterisation of tissue-specific circular RNAs in the human and mouse genomes (including 26 different tissues)	[125]
CirclncRNAnet	http://app.cgu.edu.tw/circlnc/	An integrated web-based resource for mapping functional networks of long or circular forms of noncoding RNAs	[126]
CircFunBase	http://bis.zju.edu.cn/CircFunBase/	A database for functional circular RNAs (experimentally validated and computationally predicted functions and circRNA-miRNA interaction networks)	[117]
**CircRNA databases related to human diseases**			
CSCD	http://gb.whu.edu.cn/CSCD	Comprehensive cancer-specific circRNA database	[118]
Circ2Traits	http://gyanxet-beta.com/circdb/	Comprehensive knowledgebase of potential associations of circular RNAs with human diseases (including circRNAs with disease-miRNA and SNPs-circRNA loci)	[119]
ExoRBase	http://www.exoRBase.org	Database of circRNA, lncRNA and mRNA in human blood exosomes	[104]
CircRNA disease	http://cgga.org.cn:9091/circRNADisease/	A manually curated database of experimentally supported circRNA-disease associations (48 kinds of disease)	[120]
CircR2Disease	http://bioinfo.snnu.edu.cn/CircR2Disease/	A manually curated database for experimentally supported circular RNAs associated with various diseases (725 cases between 661 circRNAs and 100 human diseases)	[121]
SomamiR 2.0	http://compbio.uthsc.edu/SomamiR	A database of cancer somatic mutations altering microRNA-ceRNA interactions (miRNA-circRNA)	[105]

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
