# Peer review of "The Regulatory Functions of Circular RNAs in Digestive System Cancers"

_cancers, 2020, doi:10.3390/cancers12030770_

Round 1
Reviewer 1 Report
In this review, the authors describe the progress made to date in understanding circular RNA biology as it relates to digestive system cancers and aging. The review begins with brief background on digestive system cancers generally and circular RNA generally, and then provides an account of the circular RNAs with roles in those cancers.
I will offer, first, my somewhat biased opinion that these works on functional circular RNA as it relates to cancer. In my perception, the majority of the literature is comprised of one-off discoveries that then receive little, if any, validation from independent laboratories. So, while I think it is clear that circular RNAs have functional significance generally, I have little credence for much of the growing literature of lone reports on the function of this or that circular RNA in cancer A or B. This is in no small part because I believe we are seeing a massive filtering of the body of experimentation for those positive results that rise to significance. I fear that the literature on circular RNAs suffers from what we see in much of biomedical literature, that is that "Most published research is false" (https://journals.plos.org/plosmedicine/article?id=10.1371/journal.pmed.0020124).
With that as a backdrop, I think it is extremely useful for a review to sift through the literature and offer a critique of what has been reported and identify areas of corroborated truth and other areas that need validation, but offer promise. It is also useful to present the currently reported findings in an organized and digestible format.
This review achieves the latter quite well, and presents what is reported about a variety of circular RNAs with some nice visual presentation and tables. However it does not achieve the former, as it is credulously repeats what is claimed in the literature and ends many paragraphs with hyperbolic and not well justified conclusions (e.g.: "These findings suggest that circHIPK3 is a therapeutic biomarker for GC and CRC", line 308, having provided no information about what these therapeutics might be or why circHIPK3 might be a biomarker of their efficacy). In general, while these closing sentences seem to be a sort of linguistic bridge between paragraphs, the authors add claims about the relevance of the work that seem unlikely.
Some specific suggestions follow
General:
The treatment of common circRNAs involved in digestive system cancers provides the greatest value as a synthesis of different findings on a set of known, reasonably highly expressed circular RNAs. In particular the treatment of CiRS-7 benefits from multiple citations and known function in neurologic development. Perhaps a figure showing these shared circRNAs and their mechanisms of action would add value to the paper? The majority of the circular RNAs are reported as functional in isolated reports, and many in a single cancer type. Certainly for other oncogenes and tumor suppressors this is not the case, and their relevance spills over to other organs and cancer types. While the authors focus on the literature surrounding digestive system tumors, they should provide additional information about whether or not these findings have been corroborated at all, both by other groups and in other tumor types. An absence of such evidence, to me, is fairly damning. I think the article "oversells" the evidence of relevance of each circular RNA. I am not terribly impressed with the section on the role of circular RNA in gut aging. This appears to rest on a single citation (#24) and unpublished data by the authors. Were this section more substantial, I would not have an issue with folding in a minor element of unpublished findings, but the ratio here is skewed. I would recommend removing the section or removing the unpublished findings. The introductory section is sparse on the description of the mechanism of formation of circular RNA (byproduct of backsplicing or degradation of an intronic lariat form). While this is mostly is beyond the scope of this review but should be briefly mentioned to facilitate the understanding of the reader who may be new to circRNA broadly. Not really major, but odd: I find it confusing that what is arguably the single most important paper on circular RNA of the last decade, the Memczak et al paper in Nature, is citation 105. Not my paper, and I have no dog in this fight, but that citation should be somewhere in the introduction where you first bring up the topic of circRNA as a competing endogenous RNA, as this is where it was first conclusively demonstrated and shown to have functional importance.Minor:
In general this manuscript needs thorough copy editing by a fluent English speaker. It's readable, but there are confusing areas. The following are not comprehensive. What follows are line numbers and issues.
37: "one of the master reason of cancer death" - awkward phrasing
41: "most universal" - bad word choice, say most common if that's what you mean. I don't have HCC is not universal.
42: "Moreover", wrong word.
57: "than linear RNAs" not sure what you mean to say.. perhaps "that linear RNAs possess"?
63: "Many litteratures" perhaps you should say "many reports"?
65: "Functions of circRNAs have been gradually showed" - awkward phrasing
72: "in this article" move this phrase to beginning of sentence.
84: "are identified" change to "have been identified"
124: "violently" poor word choice
194: "former researches" perhaps "former studies" or just "studies"
315: "derived cancer progression" word choice of derived
319: "obviously" I don't find this obvious
335: "explicated" word choice
357: "silence of" word choice or rephrase
378: "CircRNAs are circulating RNAs" I think mostly they are not circulating, though perhaps some do in exosomes. I think this is a word choice mistake unless I am missing something.
386: "The advancement of forward-thinking knowledge..." this sentence is a bit odd and out of place.
Citation 29: there is a & in the title of the journal that I think is a weird HTML carryover.
There are others I assuredly missed.
Author Response
Point-by-point responses to reviewer 1 comments/suggestions
Dear reviewer,
Thank you very much for reviewing our manuscript. We appreciate your comments, which were very pertinent, constructive and professional. In particular, the article you mentioned, "Most published research is false," gave us many ideas for the course of our future research. To provide comprehensive information on circRNAs in digestive system cancers, three parts have been added to the revised manuscript, including “6. The clinical significance of circRNAs in digestive system cancers”, “7. Human disease-related circRNA databases”, and “8. Techniques and methods of circRNA study in digestive system cancers”. All modifications in the revised manuscript are marked in red. The following are our point-by-point responses:
General part
Point 1: This review achieves the latter quite well, and presents what is reported about a variety of circular RNAs with some nice visual presentation and tables. However it does not achieve the former, as it is credulously repeats what is claimed in the literature and ends many paragraphs with hyperbolic and not well justified conclusions (e.g.: "These findings suggest that circHIPK3 is a therapeutic biomarker for GC and CRC", line 308, having provided no information about what these therapeutics might be or why circHIPK3 might be a biomarker of their efficacy). In general, while these closing sentences seem to be a sort of linguistic bridge between paragraphs, the authors add claims about the relevance of the work that seem unlikely.
Response: Thank you for your suggestions. To make these closing sentences accurately convey the points, “are” was revised to “could be” in line 409, and “is” was revised to “may be” in line 507. We have revised all similar descriptions in this manuscript.
Point 2: The treatment of common circRNAs involved in digestive system cancers provides the greatest value as a synthesis of different findings on a set of known, reasonably highly expressed circular RNAs. In particular the treatment of CiRS-7 benefits from multiple citations and known function in neurologic development. Perhaps a figure showing these shared circRNAs and their mechanisms of action would add value to the paper?
Response: To conveniently show these shared circRNAs (CiRS-7/Cdr1as, CircRNA_100290 and CircHIPK3), their mechanisms of action in digestive system cancers have been presented in a new figure (figure 2) in the revised manuscript.
Point 3: The majority of the circular RNAs are reported as functional in isolated reports, and many in a single cancer type. Certainly for other oncogenes and tumor suppressors this is not the case, and their relevance spills over to other organs and cancer types. While the authors focus on the literature surrounding digestive system tumors, they should provide additional information about whether or not these findings have been corroborated at all, both by other groups and in other tumor types. An absence of such evidence, to me, is fairly damning.
Response: Thank you for the constructive suggestion. Several circRNAs have been identified in other cancers, such as ciRS-7 in Alzheimer’s Disease (AD) (Zhao et al., 2016) and circHIPK3 in non-small cell lung cancer (NSCLC) (Lu et al., 2020). This evidence has been added to the revised manuscript.
Point 4: I think the article "oversells" the evidence of relevance of each circular RNA. I am not terribly impressed with the section on the role of circular RNA in gut aging. This appears to rest on a single citation (#24) and unpublished data by the authors. Were this section more substantial, I would not have an issue with folding in a minor element of unpublished findings, but the ratio here is skewed. I would recommend removing the section or removing the unpublished findings.
Response: We removed the section on the role of circular RNA in gut aging in the manuscript according to the reviewer’s comments.
Point 5: The introductory section is sparse on the description of the mechanism of formation of circular RNA (byproduct of backsplicing or degradation of an intronic lariat form). While this is mostly is beyond the scope of this review but should be briefly mentioned to facilitate the understanding of the reader who may be new to circRNA broadly.
Response: To facilitate reader understanding of the mechanism of circular RNA formation, we have added related information in the introductory section (lines 109 to 114, page 2).
Point 6: Not really major, but odd: I find it confusing that what is arguably the single most important paper on circular RNA of the last decade, the Memczak et al paper in Nature, is citation 105. Not my paper, and I have no dog in this fight, but that citation should be somewhere in the introduction where you first bring up the topic of circRNA as a competing endogenous RNA, as this is where it was first conclusively demonstrated and shown to have functional importance.
Response: “Citation 105” in the previous manuscript has been cited in the introduction (citation 17), where we first highlight the topic of circRNA as a competing endogenous RNA in the revised manuscript (line 122, page 2).
Minor part
Point 1: In general this manuscript needs thorough copy editing by a fluent English speaker. It's readable, but there are confusing areas. The following are not comprehensive. What follows are line numbers and issues.
Response: Thank you for your suggestions. This manuscript has been proofread by a professional English-speaking editor with a scientific background at American Journal Experts (7A5A-B647-48EA-B07F-CBEP).
Point 2: 37: "one of the master reason of cancer death" - awkward phrasing
Response: This sentence has been changed to “is also commonly diagnosed and is identified as the leading cause of death from cancer”(lines 37-38 in the revised manuscript).
Point 3: 41: "most universal" - bad word choice, say most common if that’s what you mean. I don’t have HCC is not universal.
Response: The phrase has been changed to “most common” (line 41 in the revised manuscript).
Point 4: 42: "Moreover", wrong word.
Response: The phrase has been changed to “In addition” (line 42 in the revised manuscript).
Point 5: 57: "than linear RNAs" not sure what you mean to say.. perhaps "that linear RNAs possess"?
Response: Line 107: "than linear RNAs" was changed to “that linear RNAs possess” (line 57 in the revised manuscript).
Point 6: 63: "Many litteratures" perhaps you should say "many reports"?
Response: "Many litteratures" was changed to "Many reports"(line 119 in the revised manuscript).
Point 7: 65: "Functions of circRNAs have been gradually showed" - awkward phrasing
Response: “the possible biological functions of circRNAs have been gradually showed” changed as "multiple possible biological functions of circRNAs have been discovered"(line 120-121 in the revised manuscript).
Point 8: 72: "in this article" move this phrase to beginning of sentence.
Response: "in this article" was moved to the beginning of the sentence(line 126-127 in the revised manuscript).
Point 9: 84: "are identified" change to "have been identified"
Response: "are identified" was changed to "have been identified"(line 157 in the revised manuscript).
Point 10: 124: "violently" poor word choice
Response: The word was changed to "significantly"(line 205 in the revised manuscript).
Point 11: 194: "former researches" perhaps "former studies" or just "studies"
Response: "former researches" was changed to “Previous studies"(line 337 in the revised manuscript).
Point 12: 315: "derived cancer progression" word choice of derived
Response: The phrase was changed to “might regulate cancer progression” (line 517 in the revised manuscript).
Point 13: 319: "obviously" I don’t find this obvious
Response: We removed the word "obviously"(line 521 in the revised manuscript).
Point 14: 335: "explicated" word choice
Response: The relevant content has been deleted.
Point 15: 357: "silence of" word choice or rephrase
Response: The relevant content has been deleted.
Point 16: 378: "CircRNAs are circulating RNAs" I think mostly they are not circulating, though perhaps some do in exosomes. I think this is a word choice mistake unless I am missing something.
Response: “CircRNAs are circulating RNAs (e.g., long noncoding RNAs (lncRNAs))” was changed to “CircRNAs are a type of endogenous RNA” (line 1020 in the revised manuscript).
Point 17: 386: "The advancement of forward-thinking knowledge..." this sentence is a bit odd and out of place.
Response: The sentence beginning "The advancement of forward-thinking knowledge..." was removed(line 1029 in the revised manuscript).
Point 18: Citation 29: there is a & in the title of the journal that I think is a weird HTML carryover.
Response: Citation 18: "&” was removed.
Point 19: There are others I assuredly missed.
Response: Thank you for the constructive comments. To address other problems, we will revise the article based on the comments from the other reviewers.
Reviewer 2 Report
The regulatory function of circular ARNs in digestive system cancers by Yuan and collaborators, is a very interesting initiative in the proposals of therapeutic approaches in digestive cancers. But there are major remarks to take into account.
Line-14: The author must mention the 2 distinct role of the
Circular RNAs
Line 29- : This review targeted the circular RNAs and the fact that its function is partially associated with spong RNAs, it is better than in the introduction to move lines
378-380 of the article and clearly clarified the possible role of spong RNAs.
The articles are well organized for the calcifications of the
roles of the circular RNAs, but the fact that it is a review of the articles, it lacks several references of the works of values. Consequently, figure -2 (line 375) appeared incomplete.
Overall, the author must include the functional diagrams in the form of networks and conclude by algorithm to better target the objective of this work and better identify the concept of therapeutic application.
Finally, this article does not reach its objective and a sub-chapter concerning the advertisement objective (therapeutic potential) is missing.
Author Response
Point-by-point responses to reviewer 2 comments/suggestions
Dear reviewer,
Thank you very much for reviewing our manuscript. We appreciate your comments, which are very pertinent, constructive and professional. To provide comprehensive information on circRNAs in digestive system cancers, three parts have been added to the revised manuscript, including “6. The clinical significance of circRNAs in digestive system cancers”, “7. Human disease-related circRNA databases”, and “8. Techniques and methods of circRNA study in digestive system cancers”. All modifications in the revised manuscript are marked in red. The following are our point-by-point responses:
Point 1: Line-14: The author must mention the 2 distinct role of the Circular RNAs.
Response: The Abstract has been modified as suggested in the revised manuscript. Four biological functions of circRNAs have been shown (lines 18-21, page 1).
Point 2: Line 29- : This review targeted the circular RNAs and the fact that its function is partially associated with spong RNAs, it is better than in the introduction to move lines 378-380 of the article and clearly clarified the possible role of spong RNAs.
Response: This part has been moved to “9. Conclusions” as suggested (line 1024-1025, page 19).
Point 3: The articles are well organized for the calcifications of the roles of the circular RNAs, but the fact that it is a review of the articles, it lacks several references of the works of values. Consequently, figure -2 (line 375) appeared incomplete.
Response: The clinical significance of circRNAs in digestive system cancers has been reported. Some indicators (ROC curve, sensitivity, and specificity) were used to analyse the association between circRNA expression patterns and the clinicopathological characteristics of patients. These indicators suggest that circRNAs may serve as novel potential biomarkers for the diagnosis and treatment of digestive system cancers. The new section “6. The clinical significance of circRNAs in digestive system cancers” has been added to the revised manuscript.
Point 4: Overall, the author must include the functional diagrams in the form of networks and conclude by algorithm to better target the objective of this work and better identify the concept of therapeutic application.
Response: To conveniently show these shared circRNAs (CiRS-7/Cdr1as, CircRNA_100290 and CircHIPK3), their mechanisms of action in digestive system cancers have been presented in a new figure (figure 2) in the revised manuscript.
Point 5: Finally, this article does not reach its objective and a sub-chapter concerning the advertisement objective (therapeutic potential) is missing.
Response: We agree with the comment that the previous manuscript was incomplete. Thus, we have added three new parts, “6. The clinical significance of circRNAs in digestive system cancers”, “7. Human disease-related circRNA databases”, and “8. Techniques and methods of circRNA study in digestive system cancers”. Moreover, the circRNAs with diagnostic value in digestive system cancers has been added in figure 3 in the revised manuscript.
Reviewer 3 Report
In this article, Yuan et al. provide an overview of recent progress in understanding the role of circular RNAs in digestive system cancers. The authors focus on dysregulation of circRNAs and circRNAs’ role as potential biomarkers and targets for diagnosis and prevention of digestive system cancers. Given the broad range of findings indicating circRNAs’ potential as biomarkers and/or therapeutic targets in cancer and other diseases, this summary helps to gain a more complete identification of circRNA subgroups specifically related to digestive track cancers.
The following points need to be addressed to improve the manuscript:
- Due to great variability and rapidly increasing discoveries of numerous circRNA isoformsfor the parental genes, using the circRNA names can be confounding. Therefore, in the Table 2, Table 3 and Table 4 the authors should, in addition to the circRNA names, add the circBase_ID number if it is specified in the referenced literature; g. “circ-DLG1/hsa_circ_0007203”, “circCDC14B/hsa_circ_104838”, “circDOCK1/hsa_circ_100721” etc.
- The review would benefit with an update from more recent literature. The authors should consider in their review at least the positions listed below.
Naeli P, Pourhanifeh MH, Karimzadeh MR, Shabaninejad Z, Movahedpour A, Tarrahimofrad H, Mirzaei HR, Bafrani HH, Savardashtaki A, Mirzaei H, Hamblin MR. Circular RNAs and gastrointestinal cancers: Epigenetic regulators with a prognostic and therapeutic role. Crit Rev Oncol Hematol. 2020 Jan;145:102854.
Zhai Z, Fu Q, Liu C, Zhang X, Jia P, Xia P, Liu P, Liao S, Qin T, Zhang H. Emerging Roles Of hsa-circ-0046600 Targeting The miR-640/HIF-1α Signalling Pathway In The Progression Of HCC. Onco Targets Ther. 2019 Nov 6;12:9291-9302.
Wang Q, Zhang Q, Sun H, Tang W, Yang L, Xu Z, Liu Z, Jin H, Cao X. Circ-TTC17 Promotes Proliferation and Migration of Esophageal Squamous Cell Carcinoma. Dig Dis Sci. 2019 Mar;64(3):751-758.
Zhang Y, Wang Q, Zhu D, Rong J, Shi W, Cao X. Up-regulation of circ-SMAD7 inhibits tumor proliferation and migration in esophageal squamous cell carcinoma. Biomed Pharmacother. 2019 Mar;111:596-601.
Fan L, Cao Q, Liu J, Zhang J, Li B. Circular RNA profiling and its potential for esophageal squamous cell cancer diagnosis and prognosis. Mol Cancer. 2019 Jan 23;18(1):16.
- The manuscript contains a number of grammatical errors that must be carefully corrected.
Author Response
Point-by-point responses to reviewer 3 comments/suggestions
Dear reviewer,
Thank you very much for reviewing our manuscript. We appreciate your comments, which are very pertinent, constructive and professional. To provide comprehensive information on circRNAs in digestive system cancers, three parts have been added to the revised manuscript, namely, “6. The clinical significance of circRNAs in digestive system cancers”, “7. Human disease-related circRNA databases”, and “8. Techniques and methods of circRNA study in digestive system cancers”. All modifications in the revised manuscript are marked in red. The following are our point-by-point responses:
Point 1: Due to great variability and rapidly increasing discoveries of numerous circRNA isoforms for the parental genes, using the circRNA names can be confounding. Therefore, in the Table 2, Table 3 and Table 4 the authors should, in addition to the circRNA names, add the circBase_ID number if it is specified in the referenced literature; g. “circ-DLG1/hsa_circ_0007203”, “circCDC14B/hsa_circ_104838”, “circDOCK1/hsa_circ_100721” etc.
Response: To fully show the circRNA names, Tables 2, 3 and 4 have been modified as suggested.
Point 2: The review would benefit with an update from more recent literature. The authors should consider in their review at least the positions listed below.
Response: Thank you for the constructive comments. In the revised manuscript, those new articles have been reviewed (line 126 on page 2, lines 203-204 and lines 207-212 on page 5, and line 356 on page 7). The corresponding content has been added in Table 2 and Table 3.
Point 3: The manuscript contains a number of grammatical errors that must be carefully corrected.
Response: Thank you for your suggestions. This manuscript has been proofread by a professional English-speaking editor with a scientific background at American Journal Experts (7A5A-B647-48EA-B07F-CBEP).
Round 2
Reviewer 1 Report
In this revision, the authors have addressed the bulk of my major concerns and made many strides in the readability of the paper. In addition, they have kindly altered the wording of in areas where I felt the claims of importance and relevance of the described findings were excessive. This paper covers the the core topic well, when it is reviewing specific findings regarding particular circular RNAs and the relevant literature.
Remaining issues (minor):
1) The introduction claims that gastric cancer is the leading cause of death from cancer. This is incorrect in the worldwide context, where lung is the leading cause (gastric is, I think, third). If the author are referring to China specifically they should say so.
2) Line 338: "Thus, overexpression ..." Doesn't need the "Thus". The sentence doesn't really relate to the one before it, and the behavior of the circRNA in gastric juice (relevant for clincial detection), probably doesn't relate to its relevance as a therapy.
3) Line 378: First off there needs to be a comma between "blood" and "saliva". Also the authors should provide a citation that circular RNAs are present in these settings. In plasma the amount of circular RNA is extremely small, and if I recall it is thought to exist in exosomes. That might bear mentioning.
4) The material in the "techniques and methods of circRNA study in digestive system cancers" section may be an overextension of the scope of the article, though I think the editors may be better suited to determine if this is the case than I am. For example, the section on "Study Models" (8.1) loosely describes a small subset of tumor models that are in no way specific to the analysis of circular RNA. If section 8 is retained, I would recommend removing part 8.1 at the very least. To my mind the remaining material is more properly put, in a brief form, in the introduction.
5) The conclusion and introduction both still mention "intestinal aging" despite these sections being removed from the paper. These references should be removed.
6) In my opinion the two paragraph conclusion feels rambling. I honestly thought that the end of the first paragraph was the end of the paper and then was surprised when there was more. The paper would be improved by tightening this part, under the editor's direction.
7)While the English language issues are much improved, additional attention by the editors would be warranted to improve readability.
Author Response
Point-by-point responses to reviewers 1 comments/suggestions
Dear reviewer,
Thank you very much for reviewing our manuscript. We appreciate your comments, which were very pertinent, constructive and professional. All modifications in the revised manuscript are marked in red. The following are our point-by-point responses:
Point 1: The introduction claims that gastric cancer is the leading cause of death from cancer. This is incorrect in the worldwide context, where lung is the leading cause (gastric is, I think, third). If the author are referring to China specifically they should say so.
Response: We agreed with your suggestion. The sentence “Furthermore, gastric cancer (GC) is also commonly diagnosed and is identified as the leading cause of death from cancer” has been modified as “Furthermore, gastric cancer (GC) is also commonly diagnosed and is identified as a cause of death from cancer”. (Page 1, lines 35-36)
2) Line 338: "Thus, overexpression ..." Doesn't need the "Thus". The sentence doesn't really relate to the one before it, and the behavior of the circRNA in gastric juice (relevant for clincial detection), probably doesn't relate to its relevance as a therapy.
Response: Thank you for the constructive suggestion. The sentence “Thus, overexpression of hsa_circ_0014717 may be a useful therapeutic approach for CRC” has been revised as “These results suggest that hsa_circ_0014717 may be a useful diagnostic marker for CRC.”(Page 12, lines 334-335)
3) Line 378: First off there needs to be a comma between "blood" and "saliva". Also the authors should provide a citation that circular RNAs are present in these settings. In plasma the amount of circular RNA is extremely small, and if I recall it is thought to exist in exosomes. That might bear mentioning.
Response: A comma has been added between "blood" and "saliva". The word “plasma” has been replaced by “exosomes” according to your suggestions. Furthermore, the citations have been provided in the revised manuscript. (Page 13, line 374)
4) The material in the "techniques and methods of circRNA study in digestive system cancers" section may be an overextension of the scope of the article, though I think the editors may be better suited to determine if this is the case than I am. For example, the section on "Study Models" (8.1) loosely describes a small subset of tumor models that are in no way specific to the analysis of circular RNA. If section 8 is retained, I would recommend removing part 8.1 at the very least. To my mind the remaining material is more properly put, in a brief form, in the introduction.
Response: The purpose of this section is to directly provide some experimental ideas and techniques to those who study the regulatory functions of circular RNAs in digestive system cancers, especially for junior researchers. Therefore, we suggest that this part could be retained. As suggested, the part 8.1 has been removed from the revised manuscript.
5) The conclusion and introduction both still mention "intestinal aging" despite these sections being removed from the paper. These references should be removed.
Response: These references have been removed in the revised manuscript.
6) In my opinion the two paragraph conclusion feels rambling. I honestly thought that the end of the first paragraph was the end of the paper and then was surprised when there was more. The paper would be improved by tightening this part, under the editor's direction.
Response: We have merged two conclusion paragraphs according to your suggestion.
7) While the English language issues are much improved, additional attention by the editors would be warranted to improve readability.
Response: We further revised the manuscript according to editor¢s advice.

Reviewer 2 Report
this work brings out an important notion which can be considered crucial on the role of circular RNA in cancers of the digestive system
Author Response
Point-by-point responses to reviewers 2 comments/suggestions
Dear reviewer,
Thank you very much for reviewing our manuscript. We appreciate your comments, which were very pertinent, constructive and professional. All modifications in the revised manuscript are marked in red. The following are our point-by-point responses:
Point 1: This work brings out an important notion which can be considered crucial on the role of circular RNA in cancers of the digestive system.
Response: Thank you very much for your positive comment. We have been further revised the manuscript according to other reviewer and editor.